# Multi-Color Enhanced Fluorescence Imaging of a Breast Cancer Cell with A Hole-Arrayed Plasmonic Chip

**DOI:** 10.3390/mi11060604

**Published:** 2020-06-22

**Authors:** Makiko Yoshida, Hinako Chida, Fukiko Kimura, Shohei Yamamura, Keiko Tawa

**Affiliations:** 1School of Science and Technology, Kwansei Gakuin University, 2-1 Gakuen, Sanda, Hyōgo 669-1337, Japan; skow0414@gmail.com (M.Y.); h.chida@kwansei.ac.jp (H.C.); 2Health Research Institute, National Institute of Advanced Industrial Science and Technology (AIST), 2217-14, Hayashi-cho, Takamatsu, Kagawa 761-0395, Japan; fukiko-kimura@aist.go.jp (F.K.); yamamura-s@aist.go.jp (S.Y.)

**Keywords:** multi-color imaging, breast cancer cells, surface plasmon, fluorescence microscopy, hole-array

## Abstract

Breast cancer cells of MDA-MB-231 express various types of membrane proteins in the cell membrane. In this study, two types of membrane proteins in MDA-MB-231 cells were observed using a plasmonic chip with an epifluorescence microscope. The targeted membrane proteins were epithelial cell adhesion molecules (EpCAMs) and epidermal growth factor receptor (EGFR), and Alexa^®^488-EGFR antibody and allophycocyanin (APC)-labeled EpCAM antibody were applied to the fluorescent detection. The plasmonic chip used in this study is composed of a two-dimensional hole-array structure, which is expected to enhance the fluorescence at different resonance wavelengths due to two kinds of grating pitches in a square side and a diagonal direction. As a result of multi-color imaging, the enhancement factor of Alexa^®^488-EGFR and APC-EpCAM was 13 ± 2 and 12 ± 2 times greater on the plasmonic chip, respectively. The excited wavelength or emission wavelength of each fluorescent agent is due to consistency with plasmon resonance wavelength in the hole-arrayed chip. The multi-color fluorescence images of breast cancer cells were improved by the hole-arrayed plasmonic chip.

## 1. Introduction

In recent years, a live-cell imaging technique combining surface plasmon resonance (SPR) with microscopy has been developed [1,2,3,4,5]. The enhanced electric field provided by SPR enables highly sensitive biosensing [6]. SPR can be divided into two main types, i.e., propagated and localized SPR [7], among which is grating-coupled SPR (GC-SPR)—belonging to the former type—with its periodic structure covered with thin metal films. GC-SPR can provide an enhanced electric field by direct coupling with incident light without a special optical system. A wavelength-sized grating substrate is called plasmonic chip [8,9,10,11,12] and the electric field enhanced on a plasmonic chip is generated by coupling incident light with the collective oscillation of free electrons (referred to as surface plasmon) on a metal’s surface [6]. Under observation using a fluorescence microscope, illumination light is irradiated to the sample from various angles passing through the objective lens. However, incident light can efficiently couple with plasmons using the grating vector at various azimuth angles [12]. Therefore, a brighter fluorescence image on the plasmonic chip is expected, compared with that on a glass slide, although an electric field enhanced by surface plasmon decays within 200 nm of the chip’s surface [6,9,13]. The resonance condition of GC-SPR is described as expected by Equations (1) and (2) [6,8,14]:(1)$$k_{spp}=nk_{ph}sin\theta \pm mk_{g}\;\left(m=0,1,2,\ldots \right)$$
(2)$$\frac{\omega }{c}\sqrt{\frac{\varepsilon_{1}\varepsilon_{2}}{\left(\varepsilon_{1}+\varepsilon_{2}\right)}}=n\left(\frac{\omega }{c}\right)sin\theta \pm m\frac{2\pi }{\Lambda }$$

In Equation (1), $k_{spp},\;k_{ph},\;\mathrm{and}\;k_{g}$ indicate the wavenumber vector of the surface plasmon, the incident light, and the grating, respectively, and m is the integer that indicates the order of a plasmon mode. In the Equation (2), ω and c indicate the angler frequency and the speed of light, respectively; $\varepsilon_{1}$ and $\varepsilon_{2}$ are complex dielectric constants of a metal and a dielectric, respectively; and θ, n, and Λ are the incident angle, the refractive index for dielectric media, and the pitch of a grating, respectively. As found from Equation (2), the SPR angle θ depends on the pitch Λ. In enhanced fluorescence detection, surface plasmon resonance contributes to both excitation enhancement and emission enhancement. Excitation enhancement refers to the application of an enhanced electric field to excite fluorophores, whereas emission enhancement is due to surface plasmon-coupled emission (SPCE), i.e., the fluorescence recoupled with surface plasmon again [15]. Enhancement by the effect of reflection interference is included in both excitation and emission enhancement.

In sensitive cell imaging, fluorescence imaging and Raman imaging have been widely studied by probes attached to cell membrane proteins [16,17,18,19]. Combining these conventional imaging techniques with photo thermal treatment and magnetic resonance imaging (MRI) has also been conducted [20,21,22]. In our previous studies, sensitive imaging of breast cancer cells of MCF-7 and MDA-MB-231 was obtained using a hole-arrayed plasmonic chip under an epifluorescence microscope [23], in which one kind of membrane protein, an epithelial cell adhesion molecule (EpCAM), was observed with APC-labeled antibody. The fluorescence microscopic imaging of cells is one of conventional simple methods but the combination between a microscope and a plasmonic chip accomplishes highly sensitive detection under the compact tool and simple operation. Furthermore, in the enhanced fluorescence image of membrane proteins included in cells, only the proteins faced on the chip surface can be observed due to the plasmon enhancement. Therefore, as an advantage of this technique, the distribution of proteins into cell membrane was easily detected. In the adsorption face of MCF-7 cell, the fluorescence intensity of EpCAM labeled with APC-antibody showed more than 10-fold on a plasmonic chip than that on a glass substrate, whereas in the adsorption face of MDA-MB231 cells, it showed 6-fold. The smaller enhancement observed in MDA-MB231 was considered to be underestimated due to too small intensity in the fluorescence image on the glass slide based on the fewer expression rate of EpCAM in MDA-MB231 compared with MCF-7 cells. Expression distribution of EpCAM, which in MDA-MB231 is difficult to be observed on the glass slide with a fluorescence microscope, can be sensitively detected with an improved plasmonic chip. Furthermore, the multi-proteins have not been simultaneously measured with a multi-color fluorescence imaging. On the other hand, multi-color fluorescence imaging of MDA-MB-231 was performed using a Bull’s eye-type plasmonic chip composed of concentric circles [24]. Bull’s eye-type plasmonic chip was found to provide larger fluorescence enhancement for single fluorescent nanospheres than that in the hole-arrayed plasmonic chip, because of the overlap of propagated plasmon mode at a center point of Bull’s eye circles. The membrane protein EpCAM and epidermal growth factor receptor (EGFR) of MDA-MB-231 were fluorescently labeled with allophycocyanin (APC)-anti-EpCAM antibody and Alexa^®^488-anti EGFR antibody, and the fluorescence enhancement of both APC and Alexa^®^488 were reported to be larger at a 400 nm pitch than at a 480 nm pitch [24].

However, fluorescence enhancement depends on the position inside Bull’s eye-type plasmonic chip. In order to eliminate a distribution of fluorescence intensity and to further improve the fluorescence enhancement in multi-color imaging, a hole-arrayed plasmonic chip is used in this study with two types of magnitude of $k_{g}$ based on a different pitch size of periodic patterns—along the square side direction $k_{gx}{}^{\left(1\right)}$ with m = 1 (pitch: 480 nm) and along the diagonal direction $k_{gx45}{}^{\left(2\right)}$ with m = 2 (pitch: 345 nm)—corresponding to two intrinsic resonance wavelengths (Figure 1).

The magnitude of $k_{gx}$ is determined by the pitch. Unlike the Bull’s eye-periodic structure having $k_{gx}$ with only identical pitch, a hole-arrayed plasmonic chip with two types of $k_{g}$ is expected to result in brighter multi-color fluorescence imaging when using two kinds of fluorescent dyes with different emission wavelength ranges. The brightness in a fluorescence image is evaluated as the fluorescence enhancement factor which is obtained from the ratio of the fluorescence intensities in the plasmonic chip to that in the coverslip. In this study, as shown in Figure 2, at first, the appropriate thickness of a thin silica layer prepared on the top of silver film is studied for obtaining a larger enhancement factor. Silica layer make a role of distance suppressing a fluorescence quench and convenient environment for cell adsorption. The appropriate distance from the silver surface has been evaluated as 20–30 nm for protein detection [25,26], and the distance for the largest enhancement factor including a reflection interference effect has been evaluated as 40–50 nm [9]. Though the plasmonic chip can enhance the excitation electric field closer to the silver surface, the silver layer can also quench the fluorescence within the Förster distance, and the electric field due to the reflection interference shows the maximum at 80–100 nm from the surface. On the other hand, the appropriate distance has not been shown for fluorescence imaging of cells. Cell is too large compared with protein and only fluorescence from fluorescent dyes in the cell membrane adsorbed to chip surface can be enhanced. The appropriate thickness of a silica layer is examined for maximizing an enhancement factor in this study. Then, the enhancement factors in multi-color fluorescence images are evaluated and compared with those values found in our previous study on a Bull’s eye-type plasmonic chip. More than 10-fold values are expected for dual wavelength ranges.

## 2. Materials and Methods

### 2.1. Fabrication of the Plasmonic Chip

A replica of a periodic structure was fabricated by the UV-nanoimprint method. A UV-curable resin (PAK-02-A; Toyo Gosei, Tokyo, Japan) was dropped on a 25 × 25 mm^2^ coverslip and was exposed to UV light after layering a mold, in which the center of the hole-arrayed pattern at 4 × 4 mm^2^ was fabricated. The replica was coated with thin layers of Ti, Ag, Ti, and SiO_2_ by the Rf-sputtering method. Each film thickness was < 1, 120 ± 10 < 1, and 30 or 80 nm, respectively. Two kinds of a silica layer thickness were prepared. The surface of the plasmonic chip was evaluated by atomic force microscopy (AFM; SPI3800N, SII), as shown in Figure 3a, b—which shows a pitch of 500 nm and a grove depth of 40 nm.

In order to use the plasmonic chip for cell imaging, the collagen solution was dropped on the surface of the chip. After incubation for 10 min, the solution was spread by a spin coater (1000 rpm, 30 s) and the surface was washed with MilliQ water, followed by seeding of the cells prepared in advance.

### 2.2. Cell Culture and Preparation for Microscopic Observation

The MDA-MB-231 cell line was obtained from the American Type Culture Collection (Manassas, VA, USA) and was cultured in Dulbecco’s modified Eagle medium (GIBCO, Life Technologies Co., Carlsbad, CA, USA) containing 10% fetal bovine serum harvested using trypsin. EGFR and EpCAM are known to be expressed on the cell membrane of MDA-MB-231 [27,28,29]; therefore, Alexa Fluor^®^488-labeled anti-human EGFR antibody (Alexa^®^488-EGFR; Ex: 495 nm, Em: 519 nm; Biolegend, San Diego, CA, USA) and APC-labeled anti-EpCAM monoclonal antibody (APC-EpCAM; Ex: 633 nm, Em: 660 nm; Biolegend, San Diego, CA, USA) were used for immunostaining of the membrane proteins EGFR and CD326 (EpCAM), respectively. The Alexa^®^488-EGFR and APC-EpCAM solutions were simultaneously added to the same cell solution in concentrations of 5.0 × 10^7^ molecules/cell and 1.0 × 10^5^ molecules/cell, respectively, and gently mixed for 30 min in a dark place. Then, the sample solution was gently centrifuged at 1500 rpm for 3 min and the supernatant was discarded. The precipitate was finally washed with the medium, and this process was repeated three times.

### 2.3. Microscopy

#### 2.3.1. Cell Observation

The cells were observed with an upright epi-illumination microscope (BX51WI; Olympus, Tokyo, Japan) with a 40× objective lens (UPLAN FLN 40×; Olympus, Tokyo, Japan). The light source used was a Hg lamp (BH2-RFL-T3; Olympus, Tokyo, Japan) and the detection camera was an electron multiplying charge-coupled device camera (EM-CCD, iXon; Andor, Belfast, UK). A green fluorescent protein (GFP) filter unit (UMGFPHQ; Olympus, Tokyo, Japan; λ_ex_ (460–480 nm) and λ_em_ (495–540 nm)) and a Cy5 filter unit (Cy5-4040C; Semrock, New York, NY, USA; λ_ex_ (605–650 nm) and λ_em_ (670–715 nm)) were used for the multi-color fluorescence imaging of Alexa^®^488-EGFR and APC-EpCAM, respectively.

#### 2.3.2. Microspectroscopic Measurement

In the microspectroscopic measurement, a reflection spectral image was observed using the upright microscope mounting EM-CCD camera (Luca-r, Andor) and a spectrometer (KGGCLP-50, Just Solution); a 2× objective lens (NA 0.06: Incident angle range between 0 and 3.4°) and a bright-field filter were installed, and a spectrometer was used at a slit width of 0.1 mm. Reflection spectral images were observed at the four corners of the hole-array pattern of the plasmonic chip, and the wavelength range was set at 450–750 nm. The exposure time was adjusted to ensure that the maximum intensity didn’t exceed 12,000 counts/s, because the fluorescence intensity is saturated over counts of 16,000.

The X- and Y-axes for a spectral image correspond to the wavelength and y-position of a view, respectively. Therefore, the reflection light intensities were integrated for the width of 100 pixels along the Y-axis inside the hole-array pattern as well as outside, i.e., the flat area, and the reflectivity was calculated by dividing the former intensities by the latter intensities.

## 3. Results

### 3.1. Appropriate SiO_2_ Layer Thickness for Fluorescence Imaging

An SiO_2_ film formed on a plasmonic chip can suppress fluorescence quenching by its silver layer. An SiO_2_ layer is conventionally prepared at 20–30 nm on a plasmonic chip. In this study, the SiO_2_ layer was prepared at a thickness of 30 nm (Figure 4a–c) and 80 nm (Figure 4d–f), and the fluorescence intensities were compared between them in order to improve the sensitivity in multi-color imaging. The fluorescence intensity was evaluated to be 1.4 ± 0.5 -fold larger at 80 nm than at 30 nm. Fluorescence quenching by a metal’s surface is a function of the distance from the metal surface as predicted from the Chance–Prock–Silbey (CPS) model [9,30], and the reflection interference effect can increase as increasing SiO_2_ film thickness up to 80 nm. In the experimental results obtained in our previous studies [9], 40–50 nm was identified as the best distance for enhancement with consideration of the molecular size. In cell observation with a fluorescence microscope, only the fluorescence of a labeled antibody adsorbed into the chip’s surface can be enhanced. The results of the fluorescence intensity at 80 nm being 1.4-fold greater than that at 30 nm is not inconsistent with the above theory and experimental results. Therefore, in this study, a plasmonic chip with an 80-nm-thick SiO_2_ layer was prepared for the improvement of fluorescence images. As found from Figure 4e,f, EpCAM and EGFR distributions in a MDA-MB231 cell were clearly observed individually with our improved plasmonic chip.

### 3.2. Multi-Color Imaging of MDA-MB-231 Cells

Bright field images (Figure 5a,d), fluorescence images of Alexa^®^488-EGFR (Figure 5b,e), and APC-EpCAM images (Figure 5c,f) taken on the plasmonic chip and the coverslip are shown in Figure 5. The fluorescence intensities *F* were individually evaluated at different five bright points selected from the center part of an identical cell for the Alexa^®^488 and APC images, and each background intensity B for Alexa488 and APC was also evaluated at five different points outside of the cell. The enhancement factor was calculated by using Equation (3):(3)$$Enhancemnt\;factor=\frac{F_{plasmon}-B_{plasmon}}{F_{glass}-B_{glass}}$$
where $F_{plasmon}$, $F_{glass}$, $B_{plasmon}$, and $B_{glass}$ refer to the florescence intensity of the plasmonic chip, the coverslip, the background of the plasmonic chip, and the coverslip. The enhancement factors were calculated from twenty data, i.e., five points around a center of a cell observed on the four different chips and mean value and standard deviation were obtained as shown in Table 1 and Figure 6.

In the hole-arrayed plasmonic chip, the mean enhancement factors of Alexa^®^488-EGFR and APC-EpCAM were 13 ± 2 and 12 ± 2, respectively, as shown in Table 1. These values were found to be improved compared to the values previously obtained in the Bull’s eye-type chip with a 400 nm pitch, 7 ± 2 and 9 ± 3, respectively [24].

EpCAM is generally known to be a little expression in MDA-MB 231, and that APC-EpCAM is difficult to be detected in MDA-MB231 on the conventional coverslip or glass slide [11,23]. However, the dark image of EpCAM on the coverslip was improved by 12-fold brighter image with a hole-arrayed plasmonic chip. The improved plasmonic chip is available to sensitively detecting expression distribution of membrane proteins with small expression rate in a cell. In this study, EpCAM and EGFR expression distributions were individually detected due to the fluorescence enhanced over 12-fold on the hole-arrayed plasmonic chip

## 4. Discussion

A Bull’s eye-type plasmonic chip has the unique $k_{g}$ with the same magnitude [24]. On the other hand, a hole-arrayed plasmonic chip has two kinds of $k_{g}$, i.e., $k_{gx}{}^{\left(1\right)}$ along the square side direction with m = 1 and $k_{gx45}{}^{\left(2\right)}$ along the diagonal direction with m = 2, which show the plasmon resonance at different wavelengths, individually. For $k_{gx}{}^{\left(1\right)}$ and $k_{gx45}{}^{\left(2\right)}$, the resonance wavelengths are theoretically calculated from the resonance condition described in Equations (1) and (2). Under vertical incidence, the spectra of $k_{spp}$, $k_{gx}{}^{\left(1\right)},$ and $k_{gx45}{}^{\left(2\right)}$ were derived from Equation (1), as shown in Figure 7b. The intersections correspond to the resonance modes, i.e., $k_{gx}{}^{\left(1\right)}$ and $k_{gx45}{}^{\left(2\right)}$, and the resonance wavelengths under vertical incidence at θ=0 were obtained at 695 nm and 520 nm for $k_{gx}{}^{\left(1\right)}$ and $k_{gx45}{}^{\left(2\right)}$, respectively.

Figure 7a shows the reflection spectra measured by microspectroscopy, and plasmon dips were observed at 685–710 nm and 540 nm, which were almost consistent with the resonance wavelengths of $k_{gx}{}^{\left(1\right)}$ and $k_{gx45}{}^{\left(2\right)}$ calculated from the theoretical equations. The small difference in resonance wavelengths was due to the 80-nm-thick SiO_2_ layer—which was out of consideration in the theoretical calculation; therefore, they were assigned to $k_{gx}{}^{\left(1\right)}$ and $k_{gx45}{}^{\left(2\right)}$, respectively. The plasmonic chip was illuminated with incident light from objective lens at various angles, such as a cone shape. Even when $k_{phx\;}\neq \;k_{g},\;i.e.,$ the azimuth angle φ formed by the incident plane and $k_{g}$ is not 0, a resonance condition is also satisfied, as described in Equation (1). Resonance angles were calculated against the wavelength for Bull’s eye-type and hole-arrayed-type chips every 10° of the azimuth angle and plotted in Figure 8a–c, in the illumination angle range between 0° and 48° for 40× objective lens used in this study. The excitation and emission wavelength ranges of the GFP and Cy5 filters applied to Alexa^®^488-EGFR and APC-EpCAM are shown by green and red vertical bands in Figure 8, respectively.

The resonance angle spectra of $k_{gx}{}^{\left(1\right)}$ in the 400 nm pitch Bull’s eye-type chip are shown in Figure 8a. In the excitation wavelength range of the GFP filter, the incident angles available for plasmon coupling were limited to θ = 20–48° in a 400 nm pitch (Figure 8a), but they were improved by 10–48° for $k_{gx45}{}^{\left(2\right)}$ in the hole-arrayed plasmonic chip (Figure 8c). The excitation electric field intensity by the $k_{gx45}{}^{\left(2\right)}$ in a 500 nm pitch (Figure 8c) was larger than that by $k_{gx}{}^{\left(1\right)}$ in a 400 nm pitch (Figure 8a) due to resonance angles at small angles. In addition, the incident light available for plasmon coupling was limited to φ = 0–70 in a 400 nm pitch (Figure 8a), but were improved by 0–90 for $k_{gx45}{}^{\left(2\right)}$ in a hole-arrayed chip (Figure 8c). On the other hand, in the excitation wavelength ranges of the Cy5 filter, the azimuth angles φ were limited to 0–60° in the Bull’s eye-type chip with a 400 nm pitch (Figure 8a), but in the hole-arrayed plasmonic chip, the effective azimuth angles were improved by 0–90° by $k_{gx}{}^{\left(1\right)}$ (Figure 8b). In the emission wavelength of the Cy5 filter, a hole-arrayed chip (Figure 8b) can also use more effectively the azimuth angles compared to a 400 nm- pitch Bull’s eye-type chip (Figure 8a).

As found above, two types of $k_{gx}{}^{\left(1\right)}$ and $k_{gx45}{}^{\left(2\right)}$ efficiently enhanced the fluorescence in a wide range of wavelengths. The hole-arrayed plasmonic chip with a 500 nm pitch improved the enhancement factors in the multi-color imaging of breast cancer cells in comparison to that of the values in the Bull’s eye pattern obtained in a previous study. The pattern of the plasmonic chip including a pitch and the fluorescently-labeled particles or proteins, should be selected according to the purpose, e.g., a hole-arrayed pattern for multi-color cell imaging and a Bull’s eye pattern for multi-array immunosensing. In the plasmonic pattern with several kinds of grating vectors corresponding to different size of pitches, e.g., hole arrayed-plasmonic chip with different pitches in lateral, longitudinal, and diagonal directions, individual fluorescence can be enhanced in excitation electric field or emission recoupling [15] for more than three kinds of labeled-antibody without overlap of spectra for dyes in the range of 400–700 nm A combination of the plasmonic pattern and the fluorophore is key to the highly sensitive imaging required for single molecule imaging and kinetics measurement.

## 5. Conclusions

The quality in the enhanced multi-color fluorescence images of cells labeled with different fluorophores was improved using a hole-arrayed plasmonic chip. The enhancement factors of Alexa^®^488-EGFR and APC-EpCAM in MDA-MB-231 cells obtained with the hole-arrayed plasmonic chip were 13 ± 2 and 12 ± 2, which were higher than the values obtained in a previous study with a 400 nm-pitch Bull’s eye pattern. The improved plasmonic chip is available to sensitively detecting expression distribution of membrane proteins with small expression rate in a cell. The fluorescence wavelength ranges of both Alexa^®^488-EGFR and APC-EpCAM were efficiently coupled with plasmons by two kinds of $k_{g}$ vectors, namely, $k_{gx}{}^{\left(1\right)}$ and $k_{gx45}{}^{\left(2\right)}$, in a hole-arrayed structure. It is important that the structure of a plasmonic chip is selected according to the purpose of multi-color fluorescence imaging of cells. A plasmonic patterns with more than three kinds of grating vectors by the different size of pitch will provide multi-color fluorescence imaging of more than three kinds of biomarker.

## Figures and Tables

**Figure 1 micromachines-11-00604-f001:**
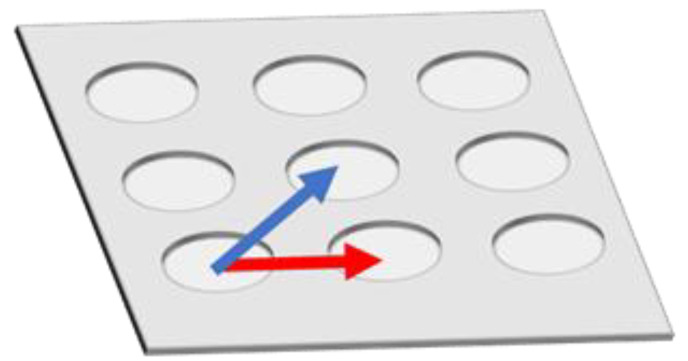
Schematic of $k_{g}$ vectors in a hole-arrayed plasmonic chip. The red and blue arrows correspond to $k_{gx}{}^{\left(1\right)}$ and $k_{gx45}{}^{\left(2\right)}$, respectively.

**Figure 2 micromachines-11-00604-f002:**
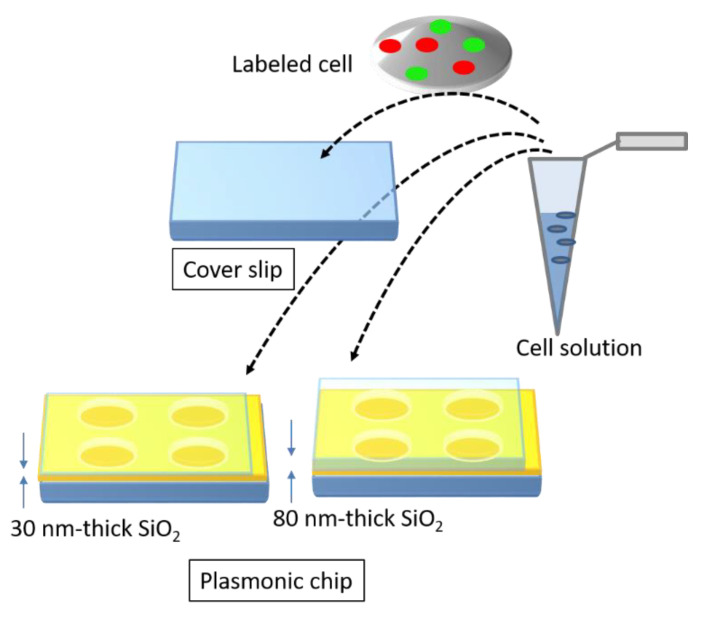
Schematic of a brief concept in this study.

**Figure 3 micromachines-11-00604-f003:**
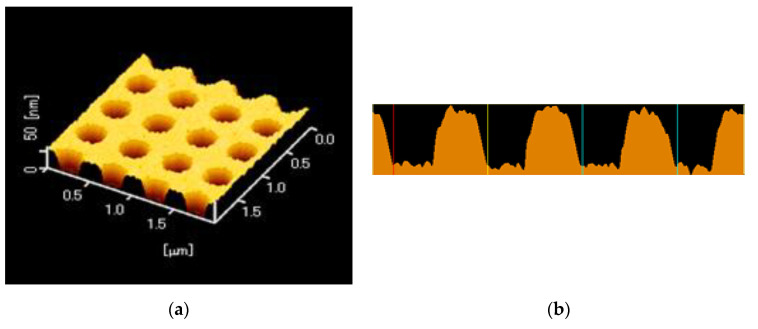
(**a**) An atomic force microscopy (AFM) image of a plasmonic chip with a hole-arrayed structure. (**b**) A cross-sectional view of the periodic structure.

**Figure 4 micromachines-11-00604-f004:**
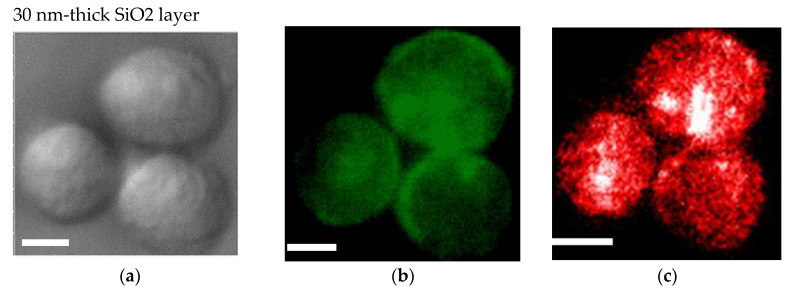
Bright field images (**a**,**d**), Alexa^®^488-EGFR fluorescence images (**b**,**e**), and allophycocyanin (APC)-epithelial cell adhesion molecule (EpCAM) fluorescence images (**c**,**f**) for MDA-MB-231 cells. The upper panels (**a**–**c**) and the lower panels (**d**–**f**) show the images taken on the plasmonic chip with a 30 nm- and a 80 nm-thick SiO_2_ layer, respectively. The max-min intensities that express the brightness contrast in fluorescence images of Alexa^®^488-EGFR and APC-EpCAM were adjusted to lie within the range of 3100 and 900 counts/s, respectively. Bars correspond to 10 μm.

**Figure 5 micromachines-11-00604-f005:**
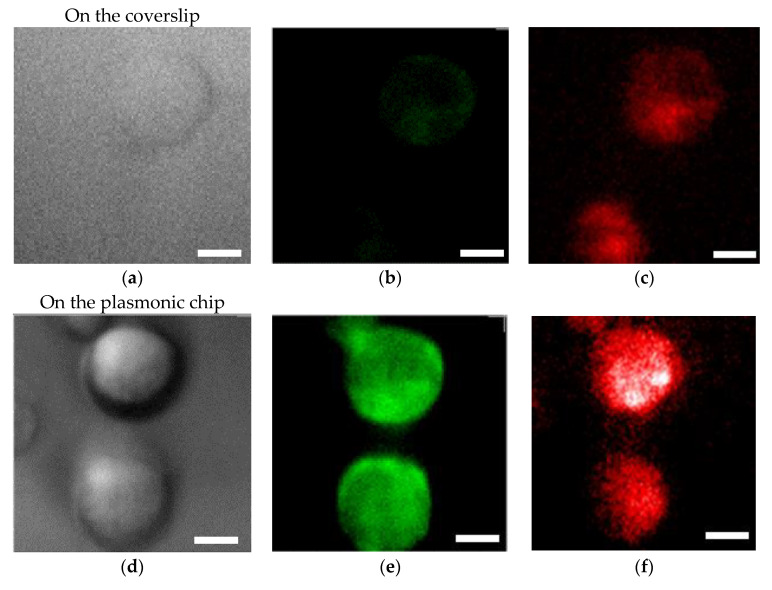
Bright field images (**a**,**d**), Alexa^®^488-EGFR fluorescence images (**b**,**e**), and APC-EpCAM images (**c**,**f**) for MDA-MB-231 cells. The upper panels (**a**–**c**) and the lower panels (**d**–**f**) show the images taken on the coverslip and plasmonic chip, respectively. The max-min intensities that express the brightness contrast in fluorescence images of Alexa^®^488-EGFR and APC-EpCAM were adjusted to lie within the range of 3400 and 1200 counts/s, respectively. Bars correspond to 10 μm.

**Figure 6 micromachines-11-00604-f006:**
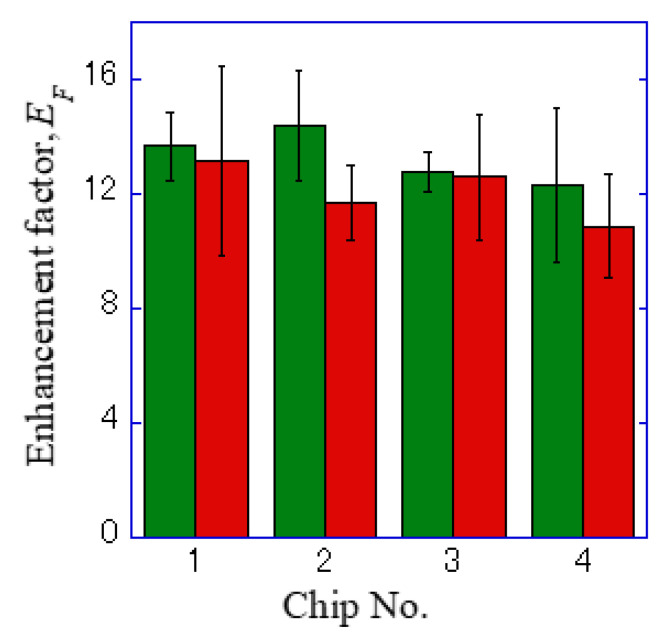
Enhancement factor for the four sheets of a plasmonic chip. The green and red bars correspond to the values for Alexa^®^488-EGFR and APC-EpCAM, respectively.

**Figure 7 micromachines-11-00604-f007:**
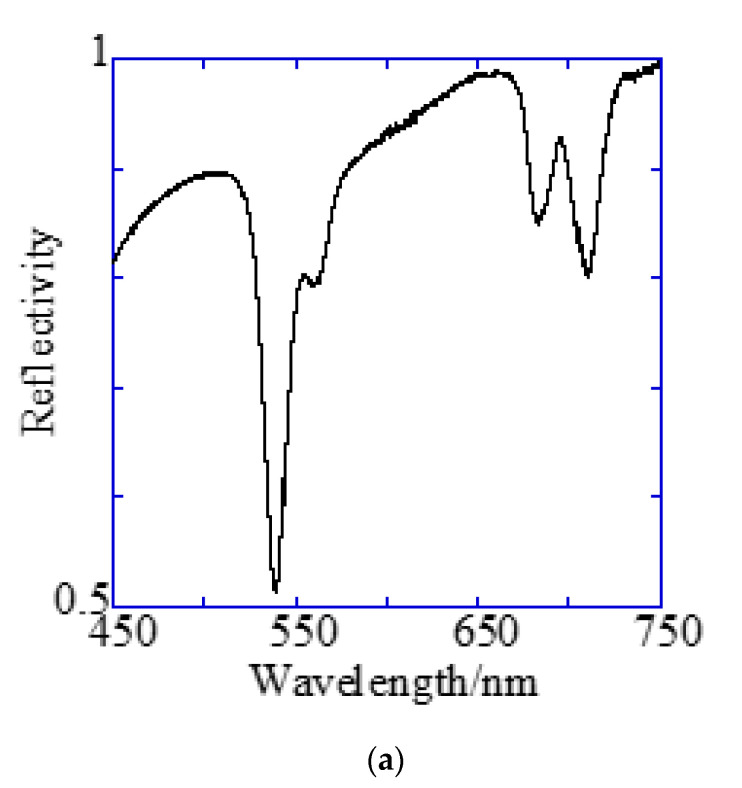
Reflection spectra and resonance wavelengths calculated from the resonance condition. (**a**) Reflection spectra measured by microspectroscopy. Two dips were confirmed at the position of the theoretically expected resonance wavelength. (**b**) The resonance wavelength at the water interface in a 500 nm pitch hole-arrayed plasmonic chip. The blue and red solid lines indicate $k_{gx}{}^{\left(1\right)}$ and $k_{gx45}{}^{\left(2\right)}$, respectively, and the intersection of the blue and red solid lines with the *k_spp_* curve are the theoretical resonance wavelength.

**Figure 8 micromachines-11-00604-f008:**
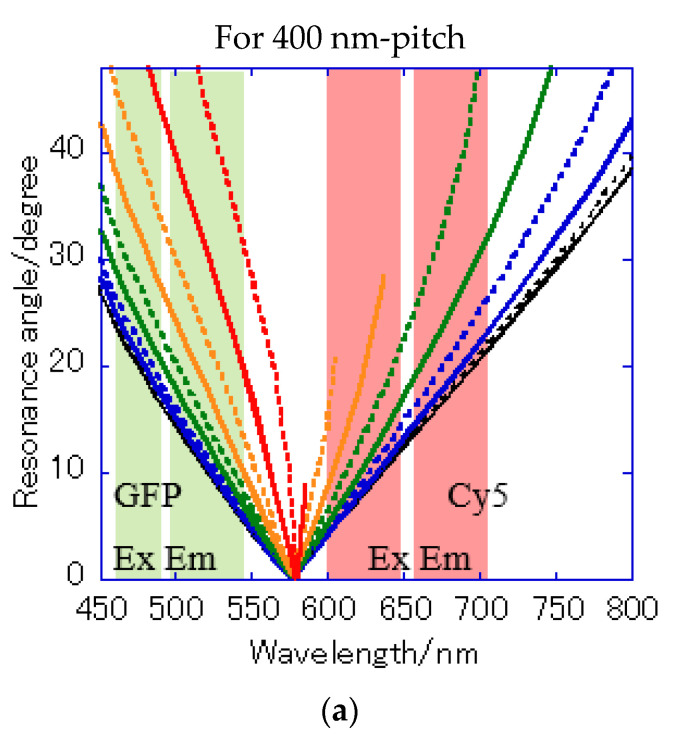
Resonance angle spectra calculated from the theoretical resonance condition at the water interface every 10° of the azimuth angle φ: (**a**) $k_{gx}{}^{\left(1\right)}$ of Bull’s eye type with 400 nm pitch; (**b**) $k_{gx}{}^{\left(1\right)}$ of hole-arrayed type with 500 nm pitch; and (**c**) $k_{gx45}{}^{\left(2\right)}$ of hole-arrayed type with 500 nm pitch. φ = 0 (black solid line), 10 (black broken line), 20 (blue solid line), 30 (blue broken line), 40 (green solid line), 50 (green broken line), 60 (orange solid line), 70 (orange broken line), 80 (red solid line), and 90 (red broken line). The green and red bands correspond to the excitation wavelength range and the emission range of the GFP and Cy5 filters, respectively.

**Table 1 micromachines-11-00604-t001:** Enhancement factor of Alexa^®^488-EGFR and APC-EpCAM.

Type	Chip No.				Average
	1	2	3	4	
Alexa^®^488-EGFR	14 ± 1	14 ± 2	13 ± 1	12 ± 3	13 ± 2
APC-EpCAM	13 ± 3	12 ± 1	13 ± 2	11 ± 2	12 ± 2

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
