# Peer review of "Multi-Color Enhanced Fluorescence Imaging of a Breast Cancer Cell with A Hole-Arrayed Plasmonic Chip"

_micromachines, 2020, doi:10.3390/mi11060604_

Round 1

Reviewer 1 Report

To whom it may concern

The manuscript Yoshida et al., in my opinion is not suitable for publication in this journal.

In this study the authors improve a technique, but they definitely do not explain advantages of using this. In the recent years many techniques have been developed for live imaging and the one proposed by the author does not represent an improvement or an important tool.

which is the relevance of this?Why one scientist would choose this technique to visualize proteins? why this could be considered better than the others already used? 

The results are poor and not well written. And the conclusion is not supporting the very few data presented and also does not underline or explain the applicability of the technique.

Author Response

To Reviewer 1

Thank you very much for a reviewer’s important suggestions and questions. We carefully considered and corrected all of them in accordance with the reviewer’s suggestions.

> The manuscript Yoshida et al., in my opinion is not suitable for publication in this journal.

> In this study the authors improve a technique, but they definitely do not explain advantages of using this. In the recent years many techniques have been developed for live imaging and the one proposed by the author does not represent an improvement or an important tool.

Thank you for a reviewer’s important suggestions.

Using the fluorescence microscopy with a plasmonic chip, only the proteins faced on the chip surface can be observed due to the plasmon enhancement. Therefore, as an advantage of this technique, the distribution of proteins into cell membrane was easily detected in addition to the merit of compact tool and simple operation.

We add the above advantage of our technique at the 63rd-68th lines in page 2.

Further, the improvement in this study is as below: Monochromatic fluorescence imaging for EGFR was accomplished with a hole-arrayed plasmonic chip in our previous study. In order to improve the fluorescence enhancement factor, Bull’s eye-type plasmonic chip in which the larger fluorescence enhancement was shown in a single nanosphere observation than that in holed-arrayed chip was used for multi-color imaging. However, the fluorescence enhancement depends on the position inside Bull’s eye-type plasmonic chip. In order to eliminate a distribution of fluorescence intensity and to further improve the fluorescence enhancement in multi-color imaging, hole-arrayed plasmonic chip was used in this study.

We add the sentences at the 69th-71st, and at the 73-75th, 80th-81th lines.

  • which is the relevance of this?Why one scientist would choose this technique to visualize proteins? why this could be considered better than the others already used? 

Thank you for important questions. As described upper, the fluorescence microscopy with a plasmonic chip can provide the information of only the proteins faced on the chip surface due to the plasmon enhancement and the distribution of proteins into cell membrane was easily detected. This is the advantages of the tool we used in addition to the compact tool and simple operation.

We show them at the 63rd-68th lines in page 2.

  • The results are poor and not well written. And the conclusion is not supporting the very few data presented and also does not underline or explain the applicability of the technique.

We are sorry for our poor construction and improved some sentences for applicability of the technique at 266th-270th lines in page 8, and some sentences were added to Introduction in order to clarify the contents presented in this paper.

Reviewer 2 Report

The manuscript entitled “Multi-Color Enhanced Fluorescence Imaging of a Breast Cancer Cell with a Hole-Arrayed Plasmonic Chip “ by Yoshida et al. describes a systematic study to determine the enhancement of the fluorescence signal measured by epifluorescence microscope. The authors used SPR as the main monitoring tool in combination with various chip surfaces including Bull’s eye type and hole-arrayed type with different pitches. The manuscript is written in a clear and engaging manner. I have several questions and comments:

  • Abstract: Please add definitions of the kinds where “two kinds of grating vectors” were mentioned. Or change wording. Here your aim should be as explanatory as possible for the non-experts.
  • Page 4 – Please add an explanation for why “The exposure time was adjusted to ensure that the maximum intensity didn’t exceed 12,000 counts/s.” what is the logic behind?
  • Figure 3 – please add labels on the image, such as label a-b-c panels as 30 nm thick SiO2 layer and d-e-f as 80 nm thick siO2 layer. Help the reader to memorize your labels.
  • Figure 2 – Also, can you add fluorescence intensity graphs next to the figures?
  • Figure 4 – please add labels by grouping the panels (similar to the comment for figure 3)
  • Table 1 – please add which locations/positions of the chips were used to sample fluorescence intensities (I guess form the dame image, but different locations, right?), where do the STDs come from? Please explain it in the text.
  • Figure 7 – please add labels by grouping the panels (similar to the comment for figure 3) also label what are the green and red bars to help the reader.
  • Critical comment: The manuscript needs a new figure 1 (concept figure), where the authors sketch the concepts used in this work. For example, sketch hole-arrayed chip (and different pitch sizes), bull-eye chip (and different pitch sizes), chips with 30 nm, and 80 nm SiO2 layers. I think at some point you also talk about an image taken from a coverslip, so please also include it, to give an overview to the reader.
  • Introduction: Please add the limit of detection for this kind of imaging, perhaps you can also add the enhancement factor to highlight the importance of the study.
  • Introduction: Please add the limitations of this kind of imaging: the distance of the substrate from the surface is the first one coming to my mind.
  • Introduction/or elsewhere: Please add your motivation to use SiO2 film.

Author Response

To Reviewer 2

Thank you very much for a reviewer’s important suggestions and questions. We carefully considered and corrected all of them in accordance with the reviewer’s suggestions.

>The manuscript entitled “Multi-Color Enhanced Fluorescence Imaging of a Breast Cancer Cell with a Hole-Arrayed Plasmonic Chip “ by Yoshida et al. describes a systematic study to determine the enhancement of the fluorescence signal measured by epifluorescence microscope. The authors used SPR as the main monitoring tool in combination with various chip surfaces including Bull’s eye type and hole-arrayed type with different pitches. The manuscript is written in a clear and engaging manner. I have several questions and comments:

  • Abstract: Please add definitions of the kinds where “two kinds of grating vectors” were mentioned. Or change wording. Here your aim should be as explanatory as possible for the non-experts.

Thank you for a suggestion. “two kinds of grating vectors” was modified to “two kinds of grating pitches in a square side and a diagonal direction” at the 19th line.

  • Page 4 – Please add an explanation for why “The exposure time was adjusted to ensure that the maximum intensity didn’t exceed 12,000 counts/s.” what is the logic behind?

At 154th line, the reason was added as “because the fluorescence intensity is saturated over counts of 16,000”.

  • Figure 3 – please add labels on the image, such as label a-b-c panels as 30 nm thick SiO2 layer and d-e-f as 80 nm thick siO2 layer. Help the reader to memorize your labels.

At the top of panels of Fig.3 a, b, c, “30 nm-thick SiO2 layer” was added and at the top of panels of Fig.3 d,e,f, “80 nm-thick SiO2 layer” was added.

  • Figure 2 – Also, can you add fluorescence intensity graphs next to the figures?

We are sorry for that we didn’t find any fluorescence intensities next to Fig.3 (revised ver.).

  • Figure 4 – please add labels by grouping the panels (similar to the comment for figure 3)

At the top of panels of Fig.4 a, b, c, “On the coverslip” was added and at the top of panels of Fig.4 d,e,f, “On the plasmonic chip” was added.

  • Table 1 – please add which locations/positions of the chips were used to sample fluorescence intensities (I guess form the dame image, but different locations, right?), where do the STDs come from? Please explain it in the text.

The analysis method was added at the 187th-188th and 194th-196th lines.

  • Figure 7 – please add labels by grouping the panels (similar to the comment for figure 3) also label what are the green and red bars to help the reader.

At the top of each panel in Figure 8, “For 400 nm-pitch” (a), “For 480 nm-pitch in the square side” (b), and “For 480 nm-pitch in the diagonal direction” (c) were added and the filter name and (Ex or Em) were inserted in the green and red bands.

  • Critical comment: The manuscript needs a new figure 1 (concept figure), where the authors sketch the concepts used in this work. For example, sketch hole-arrayed chip (and different pitch sizes), bull-eye chip (and different pitch sizes), chips with 30 nm, and 80 nm SiO2 layers. I think at some point you also talk about an image taken from a coverslip, so please also include it, to give an overview to the reader.

Thank you for an important suggestion. We added a new Figure as Figure 2 for expressing a concept in this study. The content of this study was added at the 91 st -105th lines and 113rd-114th lines.

  • Introduction: Please add the limit of detection for this kind of imaging, perhaps you can also add the enhancement factor to highlight the importance of the study.

Enhancement factor was defined at the 89th-91st lines and the value was added as “More than 10-fold values are expected for dual wavelength ranges” at the 104th-105th lines.

  • Introduction: Please add the limitations of this kind of imaging: the distance of the substrate from the surface is the first one coming to my mind.

Thank you for an important suggestion. We add the sentence of SiO2 thickness (distance from silver surface) dependence at the 91st-102nd lines.

  • Introduction/or elsewhere: Please add your motivation to use SiO2 film.

A role of silica layer was explained at the 93rd -94th lines.

Reviewer 3 Report

This paper describes a hole-arrayed plasmonic chip for sensitive imaging of fluorescence-labelled cells. The chip is an improvement of the previous plasmonic chip from the same group by providing dual-wavelength signal enhancement. The paper is well structured and easy to follow. I have the following questions for the authors:
(1) In page 2, line 58-59, the authors state that “In our previous studies, sensitive imaging of breast cancer cells of MCF-7 and MDA-MB-231 was obtained using a hole-arrayed plasmonic chip under an epifluorescence microscope”. The reference is needed here, and I also would like the authors to briefly describe the difference between the previous and the current studies.
(2) The authors state that “a hole-arrayed plasmonic chip has two kinds of wavenumber vector kg”. However, the hole array can provide many more than two different vector orientations, just like the many vectors in a 2D crystalline lattice. Please clarify this point of the number of wavenumber vectors existing in the hole-arrayed plasmonic chip.

Author Response

To Reviewer 3

Thank you very much for a reviewer’s important suggestions and questions. We carefully considered and corrected all of them in accordance with the reviewer’s suggestions.

>This paper describes a hole-arrayed plasmonic chip for sensitive imaging of fluorescence-labelled cells. The chip is an improvement of the previous plasmonic chip from the same group by providing dual-wavelength signal enhancement. The paper is well structured and easy to follow. I have the following questions for the authors:
>(1) In page 2, line 58-59, the authors state that “In our previous studies, sensitive imaging of breast cancer cells of MCF-7 and MDA-MB-231 was obtained using a hole-arrayed plasmonic chip under an epifluorescence microscope”. The reference is needed here, and I also would like the authors to briefly describe the difference between the previous and the current studies.

Thank you for a very important suggestion. We explained the motivation of multi-color imaging with a hole-arrayed plasmonic chip and difference between previous and current studies, showing results obtained in our previous studies at the 62nd-63rd and 69th-75th lines and the 80th-85th lines. The reference [23] was inserted.

>(2) The authors state that “a hole-arrayed plasmonic chip has two kinds of wavenumber vector kg”. However, the hole array can provide many more than two different vector orientations, just like the many vectors in a 2D crystalline lattice. Please clarify this point of the number of wavenumber vectors existing in the hole-arrayed plasmonic chip.

We are sorry for confusion. “two kinds of grating vector” was modified to “two kinds of grating pitches” at the 19th line, and the 83rd-87th lines. Important point is not the orientation of a grating vector but the magnitude of a grating (pitch size).

Reviewer 4 Report

This study introduces the multi-color enhanced fluorescence imaging of breast cancer cells using a hole arrayed plasmonic chip. The chip consisted of a hole-array structure with two different grating vectors that significantly enhanced the fluorescence signals. The functionality of the proposed chip was demonstrated using breast cancer cells as model analytes. This approach is useful for sensitive, multi-color fluorescence imaging of various types of cells for biomedical applications. The following comments should be addressed before the paper can be further considered:

  • In introduction, it was mentioned that “fluorescence imaging and Raman imaging have been widely studied by probes attached to cell membrane proteins. Combining these conventional imaging techniques with photo thermal treatment and magnetic resonance imaging (MRI) has also been conducted.” Could authors discuss the benefits of the approach used in this study over the existing approaches?
  • In introduction, the authors mentioned that a previous study has demonstrated sensitive imaging of breast cancer cells of MCF-7 and MDA-MB-231 using a hole-arrayed plasmonic chip under an epifluorescence microscope. As the detection signal has been significant improved using the aforementioned approach, the author should explain why further signal enhancement is required.
  • Figure 3, subfigures 3c and 3f are not clear. The background signal of 3f is too high which requires quality improvement.
  • Figure 4, subfigures 4c and 4f are not clear. Please improve their quality.
  • Some future works should be suggested. As the no. of biomarkers are limited by available fluorescence colors, what would the maximum number of biomarkers that could be detected simultaneously?

Author Response

To Reviewer 4

Thank you very much for a reviewer’s important suggestions and questions. We carefully considered and corrected all of them in accordance with the reviewer’s suggestions.

>This study introduces the multi-color enhanced fluorescence imaging of breast cancer cells using a hole arrayed plasmonic chip. The chip consisted of a hole-array structure with two different grating vectors that significantly enhanced the fluorescence signals. The functionality of the proposed chip was demonstrated using breast cancer cells as model analytes. This approach is useful for sensitive, multi-color fluorescence imaging of various types of cells for biomedical applications. The following comments should be addressed before the paper can be further considered:

  • In introduction, it was mentioned that “fluorescence imaging and Raman imaging have been widely studied by probes attached to cell membrane proteins. Combining these conventional imaging techniques with photo thermal treatment and magnetic resonance imaging (MRI) has also been conducted.” Could authors discuss the benefits of the approach used in this study over the existing approaches?

Thank you for an important question. We added the sentences at the 63rd-68th lines.

The fluorescence microscopic imaging of cells is one of conventional simple methods and the combining microscopy to a plasmonic chip accomplishes high detection sensitivity in spite of the compact tool and simple operation. Furthermore, in the enhanced fluorescence image of membrane proteins included in cells, only the proteins faced on the chip surface can be observed due to the plasmon enhancement. Therefore, as an advantage of this technique, the distribution of proteins into cell membrane was easily detected.

  • In introduction, the authors mentioned that a previous study has demonstrated sensitive imaging of breast cancer cells of MCF-7 and MDA-MB-231 using a hole-arrayed plasmonic chip under an epifluorescence microscope. As the detection signal has been significant improved using the aforementioned approach, the author should explain why further signal enhancement is required.

The purpose of this study is improvement of fluorescence enhancement factor in multi-color imaging. The sentences explaining background including previous studies were added at the 69th-71st lines, 73rd-75th lines, and 80th-82nd lines.

  • Figure 3, subfigures 3c and 3f are not clear. The background signal of 3f is too high which requires quality improvement.

The max-min intensities that express the brightness contrast in fluorescence images c and f in Figure 4 of APC-EpCAM were adjusted to lie within the range of 900 counts /s. (Figure 3 in original version changed to Figure 4 in revised version.)

  • Figure 4, subfigures 4c and 4f are not clear. Please improve their quality.

The max-min intensities that express the brightness contrast in fluorescence images c and f in Figure 5 of APC-EpCAM were adjusted to lie within the range of 1200 counts /s. (Figure 4 in original version changed to Figure 5 in revised version.)

  • Some future works should be suggested. As the no. of biomarkers are limited by available fluorescence colors, what would the maximum number of biomarkers that could be detected simultaneously?

The maximum number of biomarker detectable simultaneously depends on the feature of spectra for a dye, but more than three biomarkers may be detected. The sentences were added at the 266th-270th lines and 308th-310th lines.

Round 2

Reviewer 1 Report

I'm sorry but I'm not satisfied by the authors effort. I'm still convinced that this manuscript is not suitable for publication on Micromachines.

I think that even is definitely important to develop new techniques, it is more important to give biological relevance to the research and also a real applicability. Otherwise it is very difficult to understand and appreciate the aim of the study.

Author Response

To Reviewer 1

Thank you very much for a reviewer’s important suggestion. We corrected in accordance with the reviewer’s suggestions.

For the point of biological relevance a reviewer 1 suggested;

At lines 70-77, we added the sentence for our detailed background of membrane protein detection in breast cancer cells with a plasmonic chip as described below.

In the adsorption face of MCF-7 cell, the fluorescence intensity of EpCAM labeled with APC-antibody showed more than 10-fold on a plasmonic chip than that on a glass substrate, whereas in the adsorption face of MDA-MB231 cells, it showed 6-fold. The smaller enhancement observed in MDA-MB231 was considered to be underestimated due to too small intensity in the fluorescence image on the glass slide based on the fewer expression rate of EpCAM in MDA-MB231 compared with MCF-7 cells. Expression distribution of EpCAM, which in MDA-MB231 is difficult to be observed on the glass slide with a fluorescence microscope, can be sensitively detected with an improved plasmonic chip.

At lines 192-194, and 215-218, some sentences were added to summarize the improvement in this study as described below.

As found from Figure 4e and f, EpCAM and EGFR distributions in a MDA-MB231 cell were clearly observed individually with our improved plasmonic chip.    

The improved plasmonic chip is available to sensitively detecting expression distribution of membrane proteins with small expression rate in a cell. In this study, EpCAM and EGFR expression distributions were individually detected due to the fluorescence enhanced over 12-fold on the hole-arrayed plasmonic chip

Reviewer 4 Report

The authors have addressed my comments, so I recommend acceptance of this manuscript.

Author Response

Thank you very much for a kind review. We checked our English as possible as carefully and corrected.